# The Role of Immunosuppression for Recurrent Cholangiocellular Carcinoma after Liver Transplantation

**DOI:** 10.3390/cancers14122890

**Published:** 2022-06-11

**Authors:** Safak Gül-Klein, Paulina Schmitz, Wenzel Schöning, Robert Öllinger, Georg Lurje, Sven Jonas, Deniz Uluk, Uwe Pelzer, Frank Tacke, Moritz Schmelzle, Johann Pratschke, Ramin Raul Ossami Saidy, Dennis Eurich

**Affiliations:** 1Department of Surgery, Charité—Universitätsmedizin Berlin, Corporate Member of Freie Universität Berlin and Humboldt Universität zu Berlin, Augustenburger Platz 1, 13353 Berlin, Germany; paulina.schmitz@charite.de (P.S.); wenzel.schoening@charite.de (W.S.); robert.oellinger@charite.de (R.Ö.); georg.lurje@charite.de (G.L.); sven.jonas@charite.de (S.J.); deniz.uluk@charite.de (D.U.); moritz.schmelzle@charite.de (M.S.); johann.pratschke@charite.de (J.P.); ramin-raul.ossami-saidy@charite.de (R.R.O.S.); dennis.eurich@charite.de (D.E.); 2Department of Hematology, Oncology and Cancer Immunology, Charité—Universitätsmedizin Berlin, Corporate Member of Freie Universität Berlin and Humboldt Universität zu Berlin, Charitéplatz 1, 10117 Berlin, Germany; uwe.pelzer@charite.de; 3Department of Hepatology and Gastroenterology, Charité—Universitätsmedizin Berlin, Corporate Member of Freie Universität Berlin and Humboldt Universität zu Berlin, Augustenburger Platz 1, 13353 Berlin, Germany; frank.tacke@charite.de

**Keywords:** cholangiocellular carcinoma, liver transplantation, recurrent cholangiocellular carcinoma, reduced immunosuppression

## Abstract

**Simple Summary:**

Oncological follow-up after liver transplantation for cholangiocellular carcinoma must consider the risk of recurrence. Immunosuppressive medication with so-called mTOR inhibitors seems to have a tumor-suppressive effect, as improved survival has been shown under this medication for patients with recurrent hepatocellular carcinoma after liver transplantation. The aim of our study was to investigate recurrence and survival in relation to tumor type and type of immunosuppression for cholangiocellular carcinoma after liver transplantation. The time from liver transplantation to recurrence and survival after cancer recurrence were endpoints of the study. Significant improvement in survival for recurrent cholangiocellular carcinoma was seen after surgical resection and reduction in immunosuppression, while N1 status at transplantation and histological Grading >1 were associated with worse outcomes.

**Abstract:**

Liver transplantation (LT) for cholangiocarcinoma (CCA), or biliary tract cancer (BTC), remains controversial regarding high recurrence rates and poor prognosis. Oncological follow-up may benefit from tumor-inhibiting properties of mTOR inhibitors (mTORI), shown with improved survival for recurrent hepatocellular carcinoma (HCC) patients after LT. The aim of this study was to investigate the recurrence and survival in relation to tumor type and type of immunosuppression (IS). LT patients with CCA or mixed HCC/CCA (mHCC/CCA) (*n* = 67) were retrospectively analyzed. Endpoints were the time from LT to recurrence (*n* = 44) and survival after recurrence. Statistically significant impairment in survival for recurrent CCA (rCCA) was shown in patients not eligible for surgical resection (HR 2.46 (CI: 1.2–5.1; *p* = 0.02). Histological proven grading >1 and N1 status at initial transplantation were associated with impaired survival (HR 0.13 (CI: 0.03–0.58); *p <* 0.01 and HR 3.4 (CI: 1.0–11.65); *p =* 0.05). Reduced IS after tumor recurrence improved survival (HR 4.2/CI: 1.3–13.6; *p* = 0.02). MTORI initiation before recurrence or after had no significant impact on survival. Our data thereby indicate, similar to findings in recurrent HCC after LT, that patients with rCCA after LT benefit from a reduction in IS upon recurrence.

## 1. Introduction

Cholangiocarcinoma (CCA) is the second most common primary liver malignancy after hepatocellular carcinoma (HCC) [1]. There is a distinction between intrahepatic cholangiocarcinomas (iCCAs), originating from the intrahepatic bile ducts proximal to the second-order bile duct division, and extrahepatic cholangiocarcinomas (eCCAs). A large proportion of the CCAs are eCCA, which are divided into perihilar CCA (pCCA, Klatskin tumors) and distal CCA (pCCA), and about 20% are iCCA [2]. The incidence of iCCA has increased worldwide in all age and ethnic groups over the past few decades [3,4,5]. Coincident with the rising incidence has been an increase in iCCA-associated mortality [6]. The origin of iCCA can be either within the cirrhotic or non-cirrhotic liver. CCA accounts for approximately 3% of all tumors of gastrointestinal tumors [7,8].

Both primary hepatic malignancies, CCA, as well as hepatocellular carcinoma (HCC), are potential indications for liver transplantation (LT) under certain conditions. LT for early-stage HCC, particularly within the so-called Milan criteria, shows excellent treatment results [9]. LT may be a treatment option for unresectable perihilar CCA in combination with neoadjuvant chemotherapy and for intrahepatic CCA, as it has the best overall survival rate, but it is still controversial because of the increased recurrence rate [8,10]. In addition, mixed hepatocellular and cholangiocellular carcinoma (mHCC/CCA) are differentiated into three types according to Goodman et al. and present histological features of both tumor types [11,12]. The outcome after LT for mHCC/CCA is worse and, according to the literature, is negatively influenced in particular by the high recurrence rate [10,13,14]. Immunosuppressive therapy after LT has a special significance in the treatment of patients with primary hepatic malignoma. Thus, the proven tumor-promoting properties of calcineurin inhibitors (CNIs) must be taken into account in the specific oncological follow-up of liver-transplanted patients [15,16]. The mechanistic target of rapamycin inhibitors (mTOR inhibitors, (mTORIs)) such as sirolimus and everolimus inhibit angiogenesis in animal models and show antiproliferative properties [17,18]. Sirolimus has been shown to have slower HCC growth in vitro and longer disease-free survival [16,19]. On the other hand, a significant association was shown between cumulative CNI dose and malignancy incidence after LT, cumulative exposure to tacrolimus, and incidence of cancer after LT [20]. Recently, a significant effect of CNI reduction after diagnosis of HCC recurrence on survival was demonstrated by our group [21]. Overall, immunosuppression (IS) is a balancing act and continues to be debated in the setting of underlying oncologic disease.

The recurrence rate of the aforementioned tumors after LT is considerable, which confronts the clinician with the question of how far the extent of IS is related to the occurrence of tumor recurrence and the course after tumor recurrence, and whether improved management of immunosuppression may prolong the already limited survival in case of recurrence. Little is known about this option, so the aim of the present study was to fill this gap and derive a practical recommendation using a representative cohort of transplant patients from the context of CCA based on the experience of three decades of LT.

## 2. Materials and Methods

All patients that underwent LT for histological proven CCA or mHCC/CCA at our institution between 1988 and 2022 were included in this retrospective study. The follow-up regimen of all patients was performed in a life-long manner at our outpatient center and in correspondence with other treating physicians. Intervals of clinical examinations and laboratory assessment, as well as ultrasounds of transplant and radiological imaging (i.e., computed tomography (CT) and magnetic resonance imaging (MRI) were dependent on time after LT and recent guidelines of surveillance after malignancies; internal standard protocol required visitation twice a week to every twelve weeks. Ultrasound-guided, transcostal needle biopsies of the graft were performed at 1, 3, 5, 7, 10, and 13 years and are ongoing, as well as on individual patients’ clinical course. Diagnosis of recurrence of initial tumor disease was established by experienced radiologists and confirmed by histopathology if needed, and staging was conducted according to guidelines. TNM classification and classification of the Union for International Cancer Control (UICC) were used for staging. The grading of mHCC/CCA was determined by the histopathological report of the explanted liver. Early-onset recurrence was defined as an occurrence within two years after LT [22]. The oncological regimen was categorized into surgical or palliative concept and best supportive care (BSC). Scoring of IS was performed using the unit scale described by Vasudev et al. to enable comparison of different immunosuppressive substances (one unit for dosage of the following substances per day: 5 mg prednisone; 100 mg azathioprine; 100 mg cyclosporine; 2 mg tacrolimus; 500 mg mycophenolate mofetil; 2 mg mTORI [23]). Management of IS after diagnosis of recurrence was grouped into two categories for analysis: (i) maintaining IS or (ii) new restrictive immunosuppressive management (RIM); here, documented dose reduction or complete discontinuation of IS (CNI, MMF, GC) after diagnosis was required. Handling of mTORI was classified differently: Initiation of mTORI without reduction in other IS was classified as (i), and only patients with reduction in other immunosuppressive substances (CNI, GC, MMF) were grouped into RIM. Cross-tables and *t*-test were used in nominal variables for normally distributed or continuous variables. For testing of non-normally distributed values, Mann–Whitney U test or Kruskal–Wallis test were chosen. For survival analysis, univariate analysis and Kaplan–Meier analysis were conducted and Breslow and log-rank tests were calculated. Correlation was tested using Pearson or Spearman test depending on variables. Multivariate and univariate Cox-regression models were used, and hazard ratio (HR) and confidence interval (CI) were calculated. Relevant variables or confounders for integration in multivariate analysis were identified by clinical experience. A *p*-value of ≤0.05 was considered significant. The statistical analysis was not designed for multiple testing, and the *p*-values were considered exploratory. All statistical analyses were performed using SPSS Statistics Version 26.0 (IBM Co., Armonk, NY, USA).

## 3. Results

Between 1988 and 2022, 67 patients underwent successful LT for CCA or mHCC/CCA at our surgical department. In total, 37 (55.2%) patients were diagnosed with extrahepatic CCA (Klatskin tumor), 17 (25.4%) with intrahepatic CCA, and 13 (19.4%) with intrahepatic mHCC/CCA. The majority of patients were male (*n* = 44/65.7%), and the median age was 54.4 (30.0–70.0) years. Overall, three patients (4.5%) underwent re-LT, two because of thrombotic events and one due to recurrence of underlying primary sclerosing cholangitis (PSC). Five patients (7.5%) received combined LT with pancreaticoduodenectomy for eCCA. There was no significant difference in the rate of recurrence in the group of patients with eCCA in dependence on transplantation modus (*p* = 0.290). Overall median survival was 37.9 (3.1–333.3) months, and at the end of follow-up, 53 (79.1%) were deceased. Subgroup analysis for different entities did not show significant differences in survival for eCCA with 36.2 (3.1–333.3) months, compared with iCCA with a median survival of 57.3 (5.2–225.4) months or mHCC/CCA, with 35.2 (4.0–290.3) months, respectively (*p* = 0.96). Similar, recurrence rate did not differ with statistical significance between entities (eCCA: 26/37; 70.3%; iCCA: 12/17; 70.6%; mHCC/CCA: 6/13; 46.2%, *p* = 0.26). An overview of patient characteristics is given in Table 1.

Out of this cohort, 44 (65.7%) patients suffered from a recurrence of the initial tumor entity, with significantly reduced overall median survival after LT of 29.5 (0.13–72.8) months vs. 144.4 (4.0–333.3) months for patients without (Breslow < 0.001, log-rank < 0.001) (Figure 1).

The median age at diagnosis was 56.0 (32.0–74.0) years. The initial classification of carcinoma according to the UICC was stage I in five (11.4%) patients. In total, 18 (40.9%) patients were in stages II and III, and 3 (6.8%) patients were in stage IV. The majority of 27 (61.4%) tumors were graded G2, but 12 (27.3%) patients were diagnosed with G3 malignancies. One histopathological analysis of explanted liver concerning tumor grading was missing. Correlation analysis did reveal a significant impact of the initial UICC stage (I/II vs. III/IV) and grading (G1 vs. G2/G3) on recurrence (*p* = 0.04 and *p* = 0.03, respectively). Grading did not show an impact on overall survival after LT (Breslow < 0.29, log-rank = 0.3), whereas the UICC stage showed this effect (Breslow < 0.04, log-rank = 0.02, respectively). However, survival after recurrence was not influenced with statistical significance by either parameter (Breslow < 0.97, log-rank = 0.84; and Breslow < 0.35, log-rank = 0.18, respectively). Correlation analysis showed significance for histopathologically proven tumor manifestation in lymph nodes (N1 status) at LT and recurrence (*p* = 0.04). Multinominal regression analysis for predictors of recurrence did not show statistical impact of tumor entity (*p* = 0.32), N status (*p* = 0.14), underlying disease (*p* = 0.29), dichotomized UICC stage (*p* = 0.68), or grading (0.6). However, N1 status was associated with significantly shorter survival of 18.4 (3.9–20.7) months vs. 40.3 (3.1–333.3) months (Breslow < 0.02, log-rank = 0.02). In multivariable regression analysis, N0 status revealed a statistically significant positive impact on the overall survival of CCA after LT (HR 3.8 (CI: 1.2–12.0); *p* = 0.02). Additionally, early recurrence (<2 years after LT) was associated with the worst overall impact on survival with HR 75.5 (CI: 9.4–610.3; *p <* 0.01), but tumor entity (*p* = 0.84), underlying liver disease (*p* = 0.33), UICC stage (*p* = 0.73), or grading (*p* = 0.39) were not (Table 2).

The median time from LT to tumor recurrence was 15.8 (0.8–128.2) months. In the majority of 29 (65.9%) cases, recurrence was within 2 years after LT and was similarly associated with shorter survival after diagnosis of 6.2 (0.53–51.5) months vs. 15.8 (0.13–72.8) months (*p* = 0.035) but did not reach statistical significance in Kaplan–Meier analysis (Breslow = 0.077, log-rank = 0.09) when compared to patients with recurrence after 2 years. Frequent localizations of recurrence were within the peritoneum (*n* = 17/38.6%) and lungs (*n* = 13/29.5). Multiorgan manifestation was observed in 32 (72.7%) patients, and in 7(15.9%), singular recurrence occurred within the transplant, but neither affected survival outcomes in Kaplan–Meier analysis (Breslow = 0.67, log-rank = 0.71 and Breslow = 0.96, log-rank = 0.88, respectively).

Analyzing the therapeutic approach, 12 (27.3) patients were eligible for surgery, with 7 (15.9%) receiving only surgery only and 5 (11.4%) needing additional chemo- or radiotherapy. In 15 (34.1%) patients, palliative chemotherapy or radiotherapy was initiated, but 17 (38.6%) patients received the best supportive care only. The outcome of survival after diagnosis was significantly affected by therapy strategies, with a median survival of 18.1 (3.6–72.8) months for patients undergoing surgery with or without chemo-/radiotherapy, 12.2 (1.0–42.5) months under chemo-/radiotherapy only and just 1.2 (0.13–25.9) months when only symptomatic therapy was possible (Breslow < 0.001, log-rank <0.001) (Figure 2).

The immunosuppressive regimen upon diagnosis of CCA recurrence mainly consisted of CNIs; in 39 (88.6%) patients, these substances were administered, and the mean tacrolimus dosage was 4.1 (±2.6) mg/d. Combination therapy was used in 27 (61.4%) patients, with either MMF (*n* = 13/29.5%), glucocorticoids (*n* = 15/34.1%) or mTORI (*n* = 10/22.7%). Four (9.1%) patients received monotherapy with mTORI. The mean IS score was 3.4 (±2.4) units. The analysis presented a significant correlation of IS score with recurrence, both when analyzing absolute scores and after grouping into low (IS score 0–1.5), medium (IS score 1.6–3.5), and high (IS > 3.6) (*p* < 0.01 and *p* < 0.01, respectively). The chi^2^ test proved significant in cross-table analysis (Figure 3).

Following the diagnosis of recurrent malignancy, the number of patients with CNI therapy was reduced to 34 (85.0%), and the mean tacrolimus dosage was 2.9 (±1.9) mg/d. Wilcoxon test revealed the statistical significance of dosage reduction in tacrolimus (*p* < 0.003). Further, reductions in the number of patients receiving MMF to 11 (25.0%) and glucocorticoids to 10 (24.4%) were documented, and only the number of patients receiving mTORI increased to 15 (34.1%).

Thus, 23 (52.3%) patients of the cohort suffering from rCCA after LT were identified who underwent a reduction in IS and, therefore, were grouped into RIM. Here, the mean IS-score was 3.9 (±2.4) units before and 2.2 (±1.7) units after diagnosis of recurrence (*p* < 0.01). The median time of survival after recurrence was 16.7 (0.53–72.8) months for this group and proved to be significantly improved, compared with the group of patients with unaltered IS with 5.3 (0.13–42.5) months (*p* = 0.01). This finding was confirmed by Kaplan–Meier analysis (Breslow = 0.02, log-rank = 0.02) (Figure 4). The effect of reduction in CNI only was evaluated in 19 patients (43.2%) but did not reach statistical significance (Breslow = 0.194, log-rank = 0.09). Subgroup analysis for tumor entity showed a singular effect of CNI reduction in the group of patients suffering from Klatskin tumors on long-term survival with a median survival of 16.0 (0.9–72.8) months vs. 4.9 (0.3–23.4) months (Breslow = 0.056, log-rank = 0.04) (Figure 5). Effect of mTORI therapy after diagnosis of recurrence did not reach statistical significance in survival analysis, with a median survival of 7.5 (0.3–72.8) months for patients under mTORI therapy and 7.3 (0.13–42.5) months for those without, but a trend toward long-term benefit was observed (Breslow = 0.38, log-rank = 0.07). No clinically significant rejection was recorded for patients with RIM.

Subgroup analysis of therapeutic approach and handling of IS did not show additional benefit of RIM for patients undergoing resection with a median survival of 19.3 (6.2–72.8) and 6.8 (3.6–23.4) (Breslow = 0.12, log-rank = 0.13). Similarly, there was no impact of RIM when groups with palliative therapy (Breslow = 0.9, log-rank = 0.74) or best supportive care (Breslow = 0.92, log-rank = 0.66) were analyzed individually.

Multivariate regression analysis using clinical variables with putative empirical effect on oncological course after diagnosis of rCCA after LT confirmed the detrimental effect of grading > G1 and tumor manifestation in lymph nodes (N1 status) for survival after diagnosis of recurrence with HR 7.7 (CI: 1.7–34.7.0; *p <* 0.01) and HR 3.4 (CI: 1.0–11.7; *p =* 0.05), respectively.

Further, statistically significant impaired survival for therapeutic regimens that did not include surgical resection (HR 2.46 (CI: 1.2–5.1; *p* = 0.02) was found. Additionally, the effect of RIM was confirmed, as patients without RIM showed impaired outcomes, with statistical significance in this analysis (HR 4.2 (CI: 1.3–13.6; *p* = 0.02). MTORI therapy did not show a significant impact, regardless of administration before or after diagnosis of recurrence (*p =* 0.89 and *p =* 0.87, respectively). Complete multivariate analysis is displayed in Table 3.

## 4. Discussion

Our study aimed to investigate the impact of rCCA on survival after LT and to what extent the clinical course in terms of survival can be influenced by changes in the immunosuppressive approach. In addition, we investigated factors that influence survival for rCCA. The results of this retrospective analysis of 67 recipients with rCCA collected at our center over a period of more than 30 years indicate that surgical resection in selected cases, as well as a specific reduction in immunosuppressive therapy (RIM) after rCCA diagnosis, performed in an individualized manner in each case, is able to significantly improve patient survival. According to our current knowledge, there are no data in the literature on this aspect. However, we were able to confirm similar findings for the association of RIM and HCC in a retrospective work published recently [21]. The results of the present analysis show better survival for rCCA after LT positively influenced by reduced IS. Furthermore, targeted or palliative surgical resection remains of utmost importance for improved survival of rCCA.

Based on our analysis, comparable results were obtained for mHCC/CCA mixed-tumor patients, compared with CCA patients, in terms of recurrence and recurrence rate, although the number of patients may be small and somehow may cause bias.

In comparison, Sapisochin et al. demonstrated that mHCC/CCA mixed tumors recurred more frequently than intrahepatic CCA, while Lee et al. reported a comparable recurrence rate between the two tumor types [8,10]. In contrast, Jaradat et al. reported longer survival for patients with HCC/CCA mixed tumors, compared with patients with HCC and CCA tumors [24]. The classification of HCC/CCA mixed tumors was not included in our analysis due to low case numbers.

We found a recurrence rate of 66% in our collective, which is much higher than in patients transplanted for HCC (around 20%) but is consistent with previous studies [25,26,27]. Current global guidelines do not consider CCC a standard indication for LT [28].

In this study, recurrence was significantly associated with higher UICC stage and N1 status at initial LT. N1 status was also associated with impaired overall survival in multivariate analysis for oncological parameters at LT, highlighting the importance of adequate staging before LT to identify eligible patients in times of organ donor shortage. Previous studies found tumor size, histological parameters, and differentiation to be associated with a high recurrence rate [29]. We did not analyze size or detailed histological findings, as the focus of our study was the course of patients after recurrence.

Our multivariate regression analysis for the impact on survival for rCCA after LT showed a statistically significant effect on the impact of RIM in our population. Since treatment options are limited in recurrence besides surgical strategy, RIM should definitely be pursued with close monitoring and in a multidisciplinary approach for rCCA. We believe that individualized IS is necessary and crucial to improve the limited survival of patients.

There are only a few studies regarding immunosuppression for LT in CCA and in mHCC/CCA patients. While several studies investigated this association in HCC patients. For example, Rodríguez-Perálvarez et al. showed a more than threefold higher risk of recurrence in the first month after LT when tacrolimus levels are >10 ng/mL. In the context of immunological activity, the first month after LT seems to play an important role in the influence of CNI reduction on relapse [16]. Similarly, Vivarelli et al. identified tacrolimus as a risk factor for HCC recurrence. Furthermore, they showed a longer relapse-free survival for sirolimus, compared with CNI therapy as well as for CNI reduction [30,31]. A meta-analysis by Grigg et al. also demonstrated longer relapse-free survival and overall survival for patients on mTORI [18]. A prospective study by Geissler et al. showed longer relapse-free survival and better overall survival for patients with sirolimus up to 5 years after LT [32]. With regard to the period after relapse, longer survival was also achieved, with a reduced IS regimen [21].

Using a scoring system for IS previously introduced by Vasudev et al., we found a striking impact on the level of IS on time and the possibility of recurrence in our cohort of patients after LT for CCA, further condensing the suspicion of a fine line between rejection prevention and disastrous side effects [23]. We believe that the findings regarding targeted, individualized treatment with mTORI and CNI-reduced medication, especially after the occurrence of a relapse, could also be relevant for patients with CCA and mHCC/CCA. Here, multicenter data analysis and prospective studies are needed.

Unlike in HCC, the importance of acute cellular rejection seems to be rather secondary in the CCA context, although a definite statement is not possible due to the rarity of the diagnosis in the present setting [33,34]. In addition, we were able to demonstrate a significantly better survival, even after a recurrence, if no lymph node metastases existed.

The limitations of our study are the retrospective study design and the historical patient cohort. Especially since these tumors will continue to be misdiagnosed as HCC before LT. There is a sustained need for research on LT for CCA and mHCC/CCA both for retrospective evaluations with larger patient cohorts and with regard to a prospective study design. Further, the historical patient cohort of 30 years did not allow for differentiated analysis of systemic antineoplastic regimens or subgroups. Additionally, individual indication for RIM was not evaluated and not standardized. Indication for LT in patients with CCA might have been inhomogeneous, and accidental findings of CCA were included in this study as well.

Although our historical cohort over a period of three decades with different therapy adjustments of IS inherits heterogenous therapeutical concepts, this circumstance can also be seen as a strength since, firstly, it is known that much higher doses of IS were administered in the past, and secondly, several therapy regimens could be identified, so that the corresponding comparison groups could thus be formed. Still, indications for LT in our patients were highly variable and, in part, time-dependent; therefore, they may not reflect current therapeutic standards.

Since the results in the past on CCA and LT were very poor, LT was seen as a formal contraindication [35,36,37,38]. The poor quality of data from available studies is often due to them being single-center studies and analyses with small patient numbers and consequently limited statistical power. In addition, some of the studies did not differentiate between CCA or the presence of liver cirrhosis [39,40]. This results in a clear heterogeneity in some study patient populations, so the conclusions must be considered in a limited way from this point of view.

The role of CNI is critical with regard to acute cellular rejection (ACR) and graft loss [41]. However, data on long-term IS with CNI and trough concentrations remain controversial [42,43]. This is associated with the low implementation of protocol biopsies and treatment of rejection according to the principle of “diagnosis ex juvantibus”. In this case, up to 32% of patients may be misdiagnosed [44]. According to the analysis of our data, if recurrence occurs, the feasibility of surgical resection and RIM are mandatory therapeutical options to be explored.

While we assessed the use of a restrictive IS (RIM) as important for the overall survival of patients after LT, after a recurrence of malignancy, further reduction inherits the ability to improve following oncological regimens and appears to be safe, as we did not see an increase in rejections or graft loss in patients receiving RIM. Thus, from our point of view, CNI reduction should always be attempted and can be seen as an oncological accompanying measure in the already bleak situation, where options regarding systemic antineoplastic regimens remain poor despite rising efforts [45,46]. Multimodal concepts and fine-tuned diagnostics seeking to identify subtypes and specific genetic aberrations have been established in recent years for CCA, which allow for differentiated approaches in these rare tumors [47].

While LT for CCA remains controversial, oncological, targeted therapies and promising prospects of improved oncological outcomes by, e.g., machine perfusion of grafts, may explore new pathways [48,49].

We did not examine the pathophysiological mechanisms of this observation, but taking into account that oncogenesis is accompanied by reduced surveillance and identification properties of the immune system, a putative pathway may include the unleashing of immune function with a subsequent enhanced ability for tumor recognition [50,51,52]. Understanding the tumor entities and their characteristics using experimental models has evolved, but targeted therapies still remain scarce [53].

Thus, we conclude that oncological follow-up after LT should switch focus from primarily avoiding rejection to allowing as much immunological stimulation as possible and, therefore, risking undetected, clinically irrelevant liver tissue damage since, otherwise, our observations regarding a correlation between a reduction in immunosuppression and an improved outcome in patients after LT do not seem to be explainable. It is emerging that a protocol for IS dosing before and after relapse plays a role in both CCA and mHCC/CCA.

## 5. Conclusions

The results of this analysis emphasize the importance of follow-up after LT in terms of earlier detection of tumor recurrence with the possibility of surgical therapy and balanced mild IS according to the rule “as little as possible, as much as necessary”, and a bold reduction in IS in case of tumor recurrence as an important oncologic adjunctive measure.

## Figures and Tables

**Figure 1 cancers-14-02890-f001:**
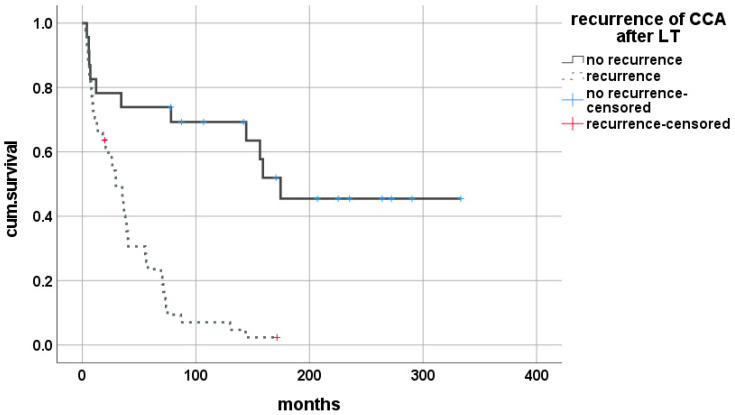
Impact of tumor recurrence on survival after LT for CCA.

**Figure 2 cancers-14-02890-f002:**
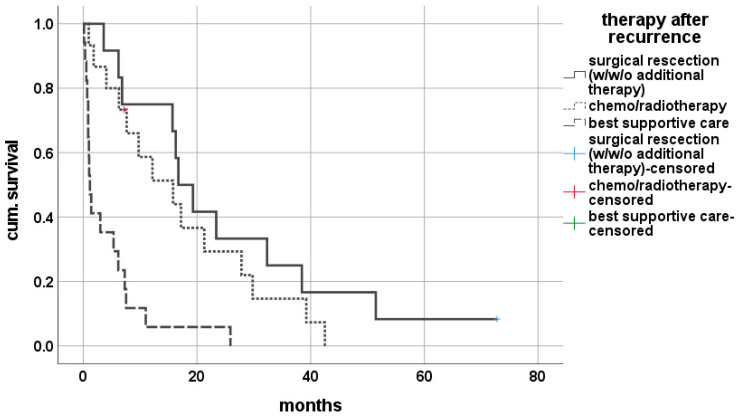
Impact of oncological strategy on survival for rCCA after liver transplantation.

**Figure 3 cancers-14-02890-f003:**
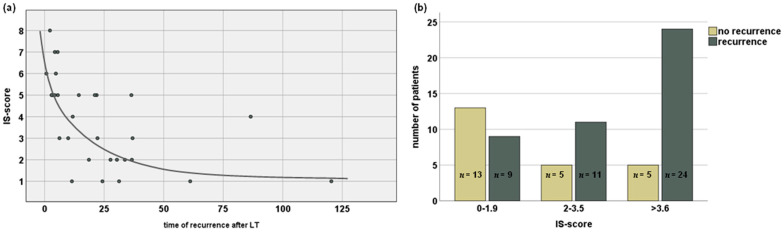
Correlation of immunosuppression score and recurrence of CCA after liver transplantation. (**a**) Score to scale immunosuppression and correlation of time to recurrence with amount of immunosuppression. (**b**) occurrence of recurrent CCA and its association with level of immunosuppression. LT—liver transplantation; IS—immunosuppression; CCA—cholangiocellular carcinoma.

**Figure 4 cancers-14-02890-f004:**
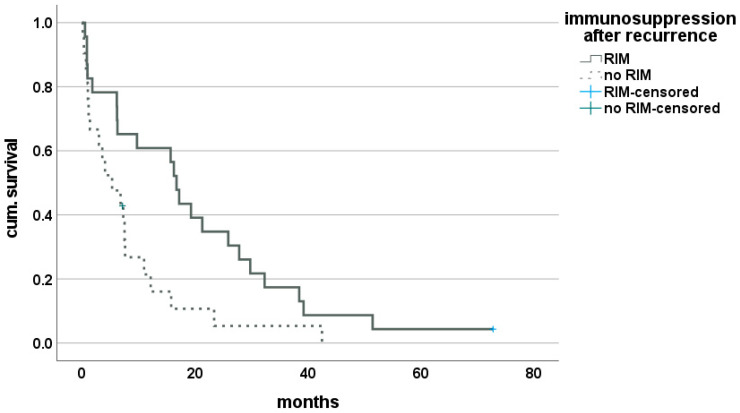
Effect of restrictive immunosuppression on survival after recurrence of CCA after LT.

**Figure 5 cancers-14-02890-f005:**
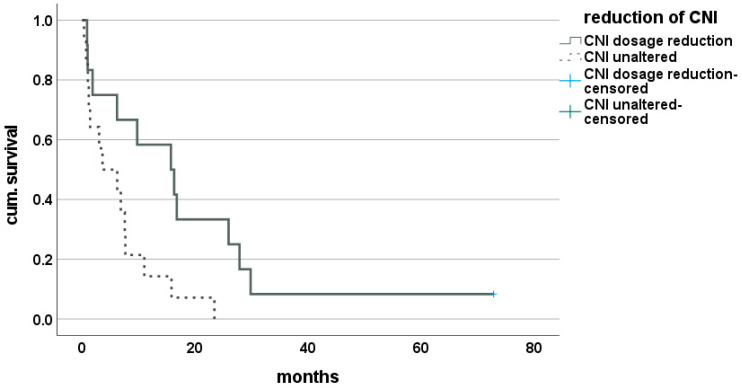
Effect of dosage reduction of CNI after LT on survival after diagnosis of with recurrence of Klatskin tumor.

**Table 1 cancers-14-02890-t001:** Characteristics for patients with and without recurrent CCA after LT.

Patients after LT for Mixed Type HCC/CCC or CCC	No Recurrence *n* = 23		Recurrence *n* = 44	
Sex (%) male female	16 (69.6) 7 (30.4)		28 (63.6) 16 (36.4)	
Entity iCCC Klatskin mixed type HCC/CCC	5 (21.8) 11 (47.8) 7 (30.4)		12 (27.3) 26 (59.1) 6 (13.6)	
Etiology of CCC (%) PSC/PBC viral hepatitis ethyltoxic unknown	7 (30.4) 2 (8.7) 5 (21.8) 9 (39.1)		12 (27.3) 3 (6.8) 6 (13.6) 23 (52.3)	
UICC stage (%) I II III IV	2 (8.7) 11 (47.8) 10 (43.5) 0 (0)		5 (11.4) 18 (40.9) 18 (40.9) 3 (6.8)	
Grading (%) G1 G2 G3 missing	5 (22.7) 13 (59.1) 4 (18.2) 1 (4.3)		5 (11.4) 27 (61.4) 12 (27.3) 0 (0)	
Re-LT (%)	2 (8.7)		1 (2.3)	
Median age at LT in years (min-max)	55.1 (30–68)		54.3 (32–70)	
Recurrence after LT (%) <2 years >2 years	- -		29 (65.9) 15 (34.1)	
Median time to recurrence in months (min–max)	-		15.8 (0.8–128.2)	
Median time of survival in months (min–max) after LT mHCC/CCA iCCA eCCA after recurrence	144.4 (4.0–333.3) 78.1 (4.0–290.3) 156.5 (77.9–225.4) 174.6 (5.7–333.3) -		29.5 (3.1–171.4) 32.3 (4.1–73.3) 32.0 (5.2–75.1) 27.7 (3.1–171.4) 7.4 (0.1–72.8)	
Oncological strategy (%) surgery chemotherapy/radiotherapy combination incl. surgery BSC	-		7 (15.9) 15 (34.1) 5 (38.6) 17 (11.4)	
IS regimen (%) CNI MMF GC mTORI CNI + others	last follow-up 20 (87.0) 5 (21.7) 2 (8.7) 5 (21.7) 10 (43.5)	prior 39 (88.6) 13 (29.5) 15 (34.1) 10 (22.7) 27 (61.4)		after 34 (85.0) 11 (25.0) 10 (24.4) 15 (34.1) 27 (61.4)
Deceased at last follow-up (%)	11 (47.8)		42 (95.5)	

LT—liver transplantation; HCC—hepatocellular carcinoma; CCA—cholangiocellular carcinoma; iCCA—intrahepatic cholangiocellular carcinoma; PSC—primary sclerosing cholangitis; PBC—primary biliary cholangitis; mHCC/CCA—mixed type hepatocellular and cholangiocellular carcinoma; CNI—calcineurin inhibitor; MMF—mycophenolat mofetil; GC—Glucocorticosteroids; mTORI—mammalian target of rapamycin inhibitor.

**Table 2 cancers-14-02890-t002:** Multivariate cox regression analysis for impact on overall survival after LT for CCA.

Parameters	*p*	Hazard Ratio	95% CI	
			Lower	Upper
Time of recurrence reference: no recurrence <2 years after LT >2 years after LT	<0.01 <0.01 0.01	- 75.54 8.11	- 9.35 1.52	- 610.31 43.38
Underlying disease	0.33	1.21	0.83	1.76
N-status at LT (N0 vs. N1) reference: N0-status	0.05	3.82	1.0	14.89
Tumor entity reference: Klatskin	0.84	0.9	0.29	2.78
Grading (G1 vs. G2/G3) reference: G1	0.39	0.57	0.16	2.08
UICC stage (I/II vs. II/IV) reference: I/II	0.73	1.23	0.37	4.10

LT—liver transplantation; N—lymph node; CCA—cholangiocellular carcinoma; iCCA—intrahepatic cholangiocellular carcinoma; UICC—Union for International Cancer Control.

**Table 3 cancers-14-02890-t003:** Multivariate cox regression analysis for impact on survival after recurrence of CCA after LT.

Parameters	*p*	Hazard Ratio	95% CI	
			Lower	Upper
Age at LT (<65/>65 years) reference: <65	0.74	1.53	0.12	20.15
Age at recurrence (<65/>65 years) reference: <65	0.66	1.5	0.25	9.01
ATG for induction of IS reference: no	0.1	3.92	0.77	19.86
Rejection before recurrence reference: yes	0.35	1.58	0.6	4.12
Time of recurrence (<2/>2 years) reference: <2	0.12	0.37	0.11	1.28
Underlying disease reference: PSC viral ethanol others	0.01 0.03 0.6 0.05	- 0.24 1.89 5.06	- 0.07 0.17 1.02	- 0.86 20.83 25.1
Tumor entity reference: Klatskin iCCA mHCC/CCA	0.1 0.22 0.86	- 2.92 0.84	- 0.53 0.12	- 16.0 5.94
Grading (G1 vs. G2/G3) reference: G1	<0.01	7.74	1.72	34.74
N-status at LT (N0 vs. N1) reference: N0-status	0.05	3.4	1.0	11.65
UICC stage (I/II vs. III/IV) reference: I/II	0.17	0.45	0.15	1.42
Surgical therapy reference: yes	0.02	2.46	1.19	5.1
mTORI before recurrence reference: yes	0.98	0.98	0.23	4.2
mTORI after recurrence reference: yes	0.87	0.91	0.28	2.91
RIM after recurrence reference: yes	0.02	4.19	1.29	13.58

LT—liver transplantation; ATG—anti-thymocyte globuline; CCA—cholangiocellular carcinoma; iCCA—intrahepatic cholangiocellular carcinoma; UICC—Union for International Cancer Control; mTORI—mammalian target if rapamycin inhibitor; RIM—restrictive immunosuppression.

## Data Availability

The data presented in this study are available on request from the corresponding author. The data are not publicly available due to the conditions of the ethics committee.

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
