# Peer review of "The Role of Immunosuppression for Recurrent Cholangiocellular Carcinoma after Liver Transplantation"

_cancers, 2022, doi:10.3390/cancers14122890_

Round 1
Reviewer 1 Report
Dear Editor, thank you so much for inviting me to revise this manuscript about biliary tract cancer.
The overall limited survival benefit provided by systemic therapies in this setting, with most patients reporting a survival rate of less than a year from the moment of diagnosis, has led to notable efforts toward the identification of novel targets and agents that could modify the natural history of these aggressive hepatobiliary malignancies. In fact, the massive use of next-generation sequencing (NGS) has led to the identification of previously unknown molecular features of CCA, including the presence of specific genetic aberrations that have been suggested to be distinctive features of iCCA and eCCA. Among these druggable alterations, fibroblast growth factor receptor (FGFR)2 gene fusions and rearrangements, isocitrate dehydrogenase-1 (IDH-1) mutations, and BRAF mutations have been widely described in CCA patients, reporting important differences between iCCA and eCCA.
Based on these premises, the paper addresses a timely topic.
The manuscript is quite well written and organized.
Tables are comprehensive and clear.
The introduction explains in a clear and coherent manner the background of this study.
We suggest the following modifications:
- Although the authors correctly included important papers in this setting, we believe the background of medical treatment for BTC should be better discussed and expanded, and the authors should include some recent papers (PMID: 32111744 ; PMID: 32824407; PMID: 32487595 ), only for a matter of consistency.
- Methods and Statistical Analysis: nothing to add.
- Discussion section: Very interesting and timely discussion. Of note, the authors should expand the Discussion section, including a more personal perspective to reflect on. For example, they could answer the following questions – in order to facilitate the understanding of this complex topic to readers: what potential does this study hold? What are the knowledge gaps and how do researchers tackle them? How do you see this area unfolding in the next 5 years? We think it would be extremely interesting for the readers.
However, we think the authors should be acknowledged for their work. In fact, they correctly addressed an important topic, the methods sound good and their discussion is well balanced.
One additional little flaw: the authors could better explain the limitations of their work, in the last part of the Discussion.
We believe this article is suitable for publication in the journal although some revisions are needed. The main strengths of this paper are that it addresses an interesting and very timely question and provides a clear answer, with some limitations.
We suggest the addition of some references for a matter of consistency. Moreover, the authors should better clarify some points.
Author Response
We want to thank the reviewer for the thoughtful revision and present the adjustments that were made based on the revision below.
--------------------------------------------------------------------------------------------------------------------------
Reviewer Point 1.
Although the authors correctly included important papers in this setting, we believe the background of medical treatment for BTC should be better discussed and expanded, and the authors should include some recent papers (PMID: 32111744 ; PMID: 32824407; PMID: 32487595 ), only for a matter of consistency.
Discussion section: Very interesting and timely discussion. Of note, the authors should expand the Discussion section, including a more personal perspective to reflect on. For example, they could answer the following questions – in order to facilitate the understanding of this complex topic to readers: what potential does this study hold? What are the knowledge gaps and how do researchers tackle them? How do you see this area unfolding in the next 5 years? We think it would be extremely interesting for the readers.
Answer:
We found the mentioned papers of high interest and relevance and added a brief paragraph into the Discussion regarding current oncological concepts as well as an outlook on therapy strategies by including the publications mentioned above such as interesting upcoming research areas were covered.
“Thus, from our point of view, CNI reduction should always be attempted and can be seen as an oncological accompanying measure in the already bleak situation, where options regarding systemic antineoplastic regimens remain poor despite rising efforts [46,47]. Multimodal concepts, fine-tuned diagnostics with effort of identification of subtypes and identification of specific genetic aberrations have been established in recent years for CCA, allowing for differentiated approaches in these rare tumors [48].
While LT for CCA remains controversial, oncological targeted therapies and promising prospects of improved oncological outcome by e.g. machine perfusion of grafts may explore new pathways [49,50].
We did not examine pathophysiological mechanisms of this observation, but taking into account, that oncogenesis is accompanied by reduced surveillance and identification properties of the immune system, a putative pathway may include the unleashing of immune function with a subsequent enhanced ability for tumor recognition [51-53]. Understanding the tumor entities and their characteristics using experimental models has evolved, but still, targeted therapies remain scarce [54].”
(Excerpt from Discussion)
--------------------------------------------------------------------------------------------------------------------------
Reviewer Point 2.
The authors could better explain the limitations of their work, in the last part of the Discussion.
Answer:
As limitations of the study must be clear and precise, we expanded this part in Discussion and hope, it now adresses all major aspects
“The limitations of our study are the retrospective study design and the historical patient cohort. Especially since these tumors will continue to be misdiagnosed as HCC before LT. There is a sustained need for research on LT for CCA and mHCC-CCA both for retrospective evaluations with larger patient cohorts and with regard to a prospective study design. Further, the historical patient cohort of 30 years did not allow for differentiated analysis of systemic antineoplastic regimens or subgroups. Also, individual indication for RIM was not evaluated and not standardized. Indication for LT in patients with CCA might have been inhomogenous and accidental findings of CCA were included in this study as well.”
(Excerpt from Discussion)

Reviewer 2 Report
I read with great interest the paper by Guel-Klein et al. regarding the risk of CCA recurrence after liver transplantation. The authors present the results of a cohort of 67 patients with CCA (mixed: intrahepatic, HCC/CCA and Klatskin), where 44 (66%) patients developed CCA recurrence after LT. Median survival was 38 months. Surgical resection and immunosuppression with mTORI improved survival, whereas early recurrence (<2 yrs) and age at recurrence (<65 yrs) were associated with a higher mortality rate.
PRO: The authors present a relatively large cohort on a topic where very scarce data are available in current literature. Their results might be helpful in the management of immunosuppression of these patients after LT.
Major limits:
- the authors show very high recurrence rate (66%) and a median survival rate of 38 months, suggesting that the graft survival rate at 5 years is way lower than 50% (futility limit). These brings to the attention how patient selection is crucial in this setting. The authors should meticulously state the indications for LT in patients with CCA (as patients in UICC stage III-IV and with N+ status were transplanted) and the eventual downstaging protocol. The main aim in this setting should be the identification of pre-LT criteria that identify patients who would most benefit from LT (and show 5-years survival rate >50%). In this view, the management of recurrence and immunosuppression after LT, and predictors of survival after recurrence are only secondary.
- inclusion of patients in a single center and from a very large period (from 1988 to 2022), when there have been made many advances in the understanding and management of CCA that may influence survival. Moreover, the number of patients in each type of CCA (extra vs intrahepatic vs HCC/CCA) is very limited to draw any conclusions in these very different categories.
- I think that statistical analysis should be revised, and that the clarity of the results should be improved. For instance:
i) I don’t understand why Pearson correlation was used to test for the association between UICC stage/Grading/IS score and recurrence.
ii) A clear multivariate analysis for the predictors of 1) overall survival (role of UICC stage? N+ status?) and 2) CCA recurrence after LT should be investigated and clearly reported. It is very difficult to understand the predictors of such outcomes throughout the text, and this is more important than the information reported in Table 2. Models investigating only pre-LT variables and pre-LT and post-LT (recurrence, surgery, immunosuppression) variables in a time-dependent analysis could be investigated.
iii) surgery supposedly improves survival, yet HR is >1 and it is non-significant in Table 2. The statement regarding age is confusing, the authors should state which group is the reference. How can recurrence <2 years and age <65 at recurrence both be risk factors for worse survival, but the hazard ratio in the first case is 0.38 and the second is 4.26?
Author Response
We want to thank the reviewer for the precise and thougtful review with many valid aspects and present our answers below in a point-by-point manner.
--------------------------------------------------------------------------------------------------------------------------
Reviewer Point 1.
The authors show very high recurrence rate (66%) and a median survival rate of 38 months, suggesting that the graft survival rate at 5 years is way lower than 50% (futility limit). These brings to the attention how patient selection is crucial in this setting. The authors should meticulously state the indications for LT in patients with CCA (as patients in UICC stage III-IV and with N+ status were transplanted) and the eventual downstaging protocol. The main aim in this setting should be the identification of pre-LT criteria that identify patients who would most benefit from LT (and show 5-years survival rate >50%). In this view, the management of recurrence and immunosuppression after LT, and predictors of survival after recurrence are only secondary.
Answer:
This is a very important aspect and many studies describe poor outcome after LT for CCA and thus, transplantation for this entitiy is not standard. This problem is of high importance, however in this study we focused the course post-LT as well as recurrence and investigated the impact of immunosuppression in this patient cohort. We acknowledged this fact in the Discussion and further highlighted the crucial need for proper patient selection in this setting.
“We found a recurrence rate of 66% in our collective, which is much higher than in patients transplanted for HCC (around 20%) but is consistent with previous studies [25-27]. Current global guidelines do not see CCC as a standard indication for LT [28].
In this study, recurrence was significantly associated with higher UICC stage and N1-status at initial LT. N1-status was also associated with impaired overall survival in multivariate analysis for oncological parameters at LT, highlighting the importance of adequate staging before LT to identify eligible patients in times of organ donor shortage. Previous studies found tumor size, histological parameters and differentiation to be associated with high recurrence rate [29]. We did not analyze size or detailed histological findings as focus of our study was course of patients after recurrence.”
(Excerpt from Discussion)
--------------------------------------------------------------------------------------------------------------------------
Reviewer Point 2
Inclusion of patients in a single center and from a very large period (from 1988 to 2022), when there have been made many advances in the understanding and management of CCA that may influence survival. Moreover, the number of patients in each type of CCA (extra vs intrahepatic vs HCC/CCA) is very limited to draw any conclusions in these very different categories.
Answer:
The reviewer mentions an important limitation, that can only be discussed but not amended in this study. We tried to further precise the presented limitations in the Discussion to meet the expected standard.
“The limitations of our study are the retrospective study design and the historical patient cohort. Especially since these tumors will continue to be misdiagnosed as HCC before LT. There is a sustained need for research on LT for CCA and mHCC-CCA both for retrospective evaluations with larger patient cohorts and with regard to a prospective study design. Further, the historical patient cohort of 30 years did not allow for differentiated analysis of systemic antineoplastic regimens or subgroups. Also, individual indication for RIM was not evaluated and not standardized. Indication for LT in patients with CCA might have been inhomogenous and accidental findings of CCA were included in this study as well.”
(Excerpt from Discussion)
--------------------------------------------------------------------------------------------------------------------------
Reviewer Point 3
I don’t understand why Pearson correlation was used to test for the association between UICC stage/Grading/IS score and recurrence.
Answer:
We apologise for the inconvenience and are very grateful for this valuable advice. A methological error was made and was revised with the analysis accordingly using biserial correlation with dichotomized variables. The new findings are presented in Results as follows:
Correlation analysis did reveal significant impact of initial UICC stage (I/II vs III/IV) and grading (G1 vs G2/G3) on recurrence (p=0.04 and p=0.03, respectively).
(Excerpt from Results)
--------------------------------------------------------------------------------------------------------------------------
Reviewer Point 4
A clear multivariate analysis for the predictors of 1) overall survival (role of UICC stage? N+ status?)
Answer:
We conducted an additional multivariable analysis on overall outcome using pre-LT parameters/variables and presented them in Results and a new Table.
“In multivariable regression analysis, N0-status revealed statistical significant positive impact on overall survival of CCA after LT (HR 3.8 (CI: 1.2–12.0; p=0.02). Also, late recurrence (>2years after LT) was associated with improved overall survival HR 0.12 (CI: 0.05–0.33; p<0.01) but tumor entity (p=0.96), underlying liver disease (p=0.07), UICC stage (p= 0.56) or Grading (p= 0.41) did not, see Table 2”
|
Parameters |
p |
Hazard Ratio |
95% CI |
|
|
lower |
upper |
|||
|
Time of recurrence (<2 vs >2 years) reference <2 years |
<0.01 |
0.12 |
0.05 |
0.33 |
|
Underlying disease |
0.71 |
1.06 |
0.80 |
1.40 |
|
N-status at LT (N0 vs N1) reference: N0-status |
0.02 |
3.78 |
1.2 |
11.96 |
|
Tumor entity reference: Klatskin iCCC mHCC/CCA |
0.96 0.83 0.98 |
- 0.87 0.98 |
- 0.26 0.32 |
- 3.0 3.05 |
|
Grading (G1 vs G2/G3) reference: G1 |
0.41 |
0.67 |
0.26 |
1.72 |
|
UICC stage (I/II vs II/IV) reference: I/II |
0.56 |
0.79 |
0.36 |
1.74 |
(Excerpt from Results)
------------------------------------------------------------------------------------------------------------------------
Reviewer Point 5
2) CCA recurrence after LT should be investigated and clearly reported. It is very difficult to understand the predictors of such outcomes throughout the text, and this is more important than the information reported in Table 2.
Answer:
This is an important information (see also above) and we conducted multinominal analysis and found no impact of the important clinical variables on recurrence. However, in univariate analysis, nodal status, grading and UICC stage were associated with recurrence showing statistical significance. These findings are all now presented in the Results as follows:
“Correlation analysis did reveal significant impact of initial UICC stage (I/II vs III/IV) and grading (G1 vs G2/G3) on recurrence (p=0.04 and p=0.03, respectively). Grading did not show impact on overall survival after LT (Breslow<0.29, Log rank 0.3), UICC stage did (Breslow<0.04, Log rank 0.02 respectively). However, survival after recurrence was not influenced with statistical significance by either parameter (Breslow<0.97, Log rank 0.84 and Breslow<0.35, Log rank 0.18 respectively). Correlation analysis showed significance for histopathological proven tumor manifestation in lymph nodes (N1-status) at LT and recurrence (p=0.04). Multinominal regression analysis for predictors of recurrence did not show statistical impact of tumor entity (p=0.32), N-status (p=0.14), underlying disease (p=0.29), dichotomized UICC stage (p=0.68) or grading (0.6). However, N1 status was associated with a significant shorter survival of 18.4 (3.9-20.7) months vs 40.3 (3.1-333.3) months (Breslow<0.02, Log rank 0.02).”
(Excerpt from Results)
--------------------------------------------------------------------------------------------------------------------------
Reviewer Point 6
Surgery supposedly improves survival, yet HR is >1 and it is non-significant in Table 2. The statement regarding age is confusing, the authors should state which group is the reference. How can recurrence <2 years and age <65 at recurrence both be risk factors for worse survival, but the hazard ratio in the first case is 0.38 and the second is 4.26?
Answer:
The reviewer adressed critical flaws in this Table and we revised the multivariate cox-regression analysis and now present references. Further, we dichotomized the therapeutical regimen (surgery-yes/no) to clarify the results. Of note, in this analysis, age did not show impact for survival and the Results and the Discussion were altered accordingly. See below for the new Table that is also presented in Results.
|
Parameters |
p |
Hazard Ratio |
95% CI |
||
|
lower |
upper |
||||
|
Age at LT (<65 / >65 years) reference: <65 |
0.74 |
1.53 |
0.12 |
20.15 |
|
|
Age at recurrence (<65 / >65 years) reference: <65 |
0.66 |
1.5 |
0.25 |
9.01 |
|
|
ATG for induction of IS reference: no |
0.1 |
3.92 |
0.77 |
19.86 |
|
|
Rejection before recurrence reference: yes |
0.35 |
1.58 |
0.6 |
4.12 |
|
|
Time of recurrence (<2/>2 years) reference: <2 |
0.12 |
0.37 |
0.11 |
1.28 |
|
|
Underlying disease reference: PSC viral ethanol others |
0.01 0.03 0.6 0.05 |
- 0.24 1.89 5.06 |
- 0.07 0.17 1.02 |
- 0.86 20.83 25.1 |
|
|
Tumor entity reference: Klatskin iCCA mHCC/CCA |
0.1 0.22 0.86 |
- 2.92 0.84 |
- 0.53 0.12 |
- 16.0 5.94 |
|
|
Grading (G1 vs G2/G3) reference: G1 |
<0.01 |
7.74 |
1.72 |
34.74 |
|
|
N-status at LT (N0 vs N1) reference: N0-status |
0.05 |
3.4 |
1.0 |
11.65 |
|
|
UICC stage (I/II vs III/IV) reference: I/II |
0.17 |
0.45 |
0.15 |
1.42 |
|
|
Surgical therapy reference: yes |
0.02 |
2.46 |
1.19 |
5.1 |
|
|
mTORI before recurrence reference: yes |
0.98 |
0.98 |
0.23 |
4.2 |
|
|
mTORI after recurrence reference: yes |
0.87 |
0.91 |
0.28 |
2.91 |
|
|
RIM after recurrence reference: yes |
0.02 |
4.19 |
1.29 |
13.58 |
|

Round 2
Reviewer 1 Report
Acceptance.
Reviewer 2 Report
I commend the authors for their responses and the revised manuscript.
One minor change: in table 2, the reference should be no recurrence, and then the HR of early vs late recurrence can be confronted; otherwise, the survival analysis in the current form is being performed only in patients with CCA recurrence and not the whole population under study.